# Conservative Management for Retained Products of Conception in Late Pregnancy

**DOI:** 10.3390/healthcare11020168

**Published:** 2023-01-05

**Authors:** Risa Fujishima, Kaoru Kawasaki, Kaori Moriuchi, Reona Shiro, Yoshie Yo, Noriomi Matsumura

**Affiliations:** Department of Obstetrics and Gynecology, Faculty of Medicine, Kindai University, 377-2 Ohno-Higashi, Osaka-Sayama 589-8511, Osaka, Japan

**Keywords:** retained products of conception (RPOC), postpartum hemorrhage, human chorionic gonadotropin (hCG)

## Abstract

This retrospective study aims to compare the early manual removal of placenta (MROP) and conservative management of retained products of conception (RPOC) after 34 weeks of gestation. Nineteen cases underwent MROP within 24 h of delivery, of which nine patients had no symptoms requiring emergent treatment. These 9 patients (group M) were compared with 22 patients who were treated conservatively (group C). Massive bleeding was observed in 5 (56%) patients in group M and 11 (50%) patients in group C, with no significant difference in frequency. However, the lowest hemoglobin level within 72 h after massive bleeding was lower in group M (median: 6.7 vs. 7.7 g/dL, *p* = 0.029), suggesting that massive bleeding occurred in a short period of time. On the other hand, a retained placenta was observed in four patients in group M after the MROP; however, the placenta disappeared more quickly than in group C (median; 1.0 vs. 99.0 days, *p* = 0.009). In group C, all bleeding and infection occurred within 60 days of delivery, including heavy bleeding in six cases during the placental-extraction trial. Human chorionic gonadotropin in group C fell below the measurable threshold at a median of 67 days postpartum. In conclusion, for RPOC without urgent symptoms, early MROP and conservative treatment have their advantages and disadvantages. Randomized controlled trials are needed to determine which of those treatments is superior.

## 1. Introduction

A delay in placental delivery may cause heavy bleeding in late pregnancy [1]. Therefore, it is recommended that obstetric providers use uterotonic agents immediately after delivering the fetus and deliver the placenta by controlled cord traction [2]. Retained products of conception (RPOC), in which placental tissue remains in the uterus after delivery, occur in about 1% of pregnant women and cause severe postpartum hemorrhage [3,4]. The National Institute for Health and Clinical Excellence (NICE) [5] and the World Health Organization (WHO) [6] recommend that obstetric providers perform manual removal of the placenta (MROP) if it is not expelled within 30 min to 1 h after the baby is delivered [7]. However, MROP may increase the risk of hysterectomy and death due to heavy bleeding. Additionally, the placenta often remains partly after MROP [8]. If conservative management is chosen immediately after delivery, or retained placenta occurs after MROP, there is a risk of subsequent massive hemorrhage or infection [9,10,11,12]. Therefore, some studies suggest that RPOC should be removed by performing surgical treatment such as transcervical resection (TCR) to avoid bleeding and infection during conservative management [13,14]. However, surgical RPOC removal also carries a high risk of massive bleeding [15].

The conservative management of RPOC is to wait and expect spontaneous expulsion without surgical treatment. Recently, several Japanese facilities have reported that RPOC spontaneously disappeared without surgical management [16,17]. However, evidence for the conservative management of RPOC, especially after term delivery, is lacking, and there are no criteria to determine which cases can achieve spontaneous resolution [18,19]. It is also unclear when major bleeding or infection occurs and how long it will take for RPOC to be absorbed spontaneously.

Our hospital is the only university hospital in the South Osaka district, and most patients with RPOC in this area are referred to our hospital. We have treated many cases with RPOC conservatively until they spontaneously disappear. This study aimed to analyze the clinical course of RPOC after 34 weeks of gestation and compare MROP and conservative management.

## 2. Materials and Methods

### 2.1. Patients

This study included 41 patients who delivered after 34 weeks of gestation and were managed for RPOC at Kindai University Hospital from January 2013 to March 2022. Ten patients underwent MROP within 24 h following delivery due to severe symptoms such as heavy bleeding and infection. The remaining 31 RPOC cases were divided into the following two groups: those who underwent MROP within 24 h of delivery without urgent symptoms (group M) and those who were managed conservatively without MROP (group C) (Figure 1). At the time of heavy bleeding, uterine artery embolization (UAE) was performed for hemostasis.

Patients with RPOC who delivered before 20 weeks of gestation and did not undergo UAE were also included to evaluate serial changes in serum human chorionic gonadotropin (hCG) levels.

This study was approved by the ethical review board of Kindai University (Approval number: R04-166).

### 2.2. Endpoint

Clinical data from patients’ electronic medical records were reviewed retrospectively. The following variables were investigated: gestational weeks and mode of delivery, RPOC diameter at diagnosis, blood flow in RPOC, bleeding, infection, UAE, MROP, TCR, dilatation and curettage (D&C), hysterectomy, blood transfusion, the onset of heavy bleeding, date of RPOC disappearance, and serum hCG levels. Ultrasound was performed in all cases. Blood flow in RPOC was detected by performing Doppler ultrasound. Contrast-enhanced computed tomography (CT) was performed to detect the bleeding point in the acute phase. Contrast-enhanced magnetic resonance imaging (MRI) was used to evaluate blood flow, volume, and myometrial invasion of RPOC in the chronic phase. We defined “heavy bleeding” as bleeding accompanied by a change in vital signs; “moderate bleeding” as normal menstrual blood loss; and “light bleeding” as minor blood loss. Spontaneous resolution was defined as RPOC disappearance without removal of the placenta, which was confirmed by ultrasound or hysteroscopy.

### 2.3. Statistical Analysis

Statistical analyses were performed using GraphPad Prism 6 (GraphPad Software, La Jolla, CA, USA). Fisher’s exact test and the Mann-Whitney *U* test were used to compare continuous variables; *p* < 0.05 was considered statistically significant.

## 3. Results

There were 41 cases of RPOC following delivery after 34 weeks of gestation, with no deaths or hysterectomies. Our institution manages RPOC cases conservatively with informed consent unless there are no signs of placental delivery or urgent conditions such as heavy bleeding. However, in such cases, if patients request, we also perform MROP within 24 h of delivery. The placenta was manually removed in 19 cases within 24 h of delivery. The remaining 22 cases were managed conservatively, without MROP.

### 3.1. Nineteen Cases with MROP within 24 Hours of Delivery

Nineteen cases were delivered vaginally, with a median (range) of 39.9 (34.7–41.1) weeks of gestation at delivery.

Overall, 10 of the 19 cases had symptoms which require emergent MROP, including 9 cases with massive hemorrhage (Cases 1–9) and 1 case with infection (Case 10) (Figure 2A). All cases required blood transfusion during MROP. One case (Case 3) had RPOC after MROP, which spontaneously resolved at day 46.

The remaining nine cases underwent MROP without any urgent symptoms (Case 11–19; group M). Five of the nine cases experienced heavy bleeding during the procedure (Cases 11–15), and four of them required blood transfusion. Two cases (Cases 11 and 12) experienced heavy bleeding again, and one case (Case 12) had infection. Four of the nine cases (Case 11, 12, 13, 16) had RPOC after MROP, which spontaneously resolved in all cases within 99 days. One case (Case 14), in which the retained placenta was completely removed by MROP, had heavy bleeding the day after MROP and underwent UAE.

### 3.2. Twenty-Two Cases with Conservative Management

Twenty-two patients were treated conservatively (group C). Overall, 3 of the 22 cases were delivered via cesarean section. The median RPOC diameter was 46 mm, with five cases exceeding 100 mm (Figure 2B). Seventeen cases had blood flow in RPOC on contrast-enhanced CT, contrast-enhanced MRI, or Doppler ultrasound (Figure 2B and Figure 3). Three cases had no blood flow, and two were not evaluable.

Twelve cases experienced heavy bleeding and nine cases required hemostasis through UAE. Overall, 6 of the 12 cases had unprovoked bleeding. The remaining six cases of major bleeding occurred during attempts to remove the placenta, including five cases (Cases 23, 29, 31, 37, 41) of pulling of the placenta, which appeared to be about to be expelled by the opened cervix, and one case (Case 30) of dilatation and curettage (D&C). Even though the contrast-enhanced MRI did not show any blood flow, Case 31 had heavy bleeding due to placental traction.

Three cases (Cases 20, 26, 28) had infection, all of which were cured with antibiotics. Case 20 underwent MROP on postpartum day 7 and showed signs of infection on postpartum day 15. Eight cases had no complications, such as bleeding or infection, and three (Case 25, 34, 38) were delivered via caesarean section. RPOC resolved spontaneously in 17 of 22 cases (77%) without surgical intervention, and the median time from delivery to resolution was 130 days. There was no difference between heavy (*n* = 12) and light (*n* = 9) bleeding cases in terms of RPOC diameter at diagnosis, blood flow on CT or MRI, and the time from delivery to resolution of RPOC (Table 1).

Regarding the timing, heavy bleeding occurred within 60 days postpartum in 22 cases with conservative management. Blood transfusion was required within 60 days postpartum, and UAE was required within 30 days postpartum. Infection occurred within 30 days postpartum (Figure 4A–D, red lines). Heavy bleeding, UAE, blood transfusion, and infection occurred within 40, 40, 20, and 30 days postpartum, respectively, in 16 cases, excluding 6 patients with hemorrhage in the placental removal trial (Figure 4A–D, blue lines).

### 3.3. Comparison between Group M and Group C

To determine the relative merit of MROP versus conservative treatment, we evaluated the cumulative incidence of events in 9 MROP cases without urgent symptoms (group M) and 22 conservative treatment cases (group C). The cumulative incidence of heavy bleeding, transfusions, UAE, and infections did not differ between these two groups (Figure 5A–D).

Heavy bleeding occurred in 5 (56%) cases in group M and 11 (50%) cases in group C, with no significant difference in frequency. The lowest hemoglobin level within 72 h after heavy bleeding was lower in group M than group C (median; 6.7 vs. 7.7 g/dL, *p* = 0.029), suggesting that massive bleeding occurred in a short period of time. There was no significant difference in the amount of blood transfusion and the lowest fibrinogen level within 72 h after heavy bleeding (Figure 6).

Retained placenta was observed in four patients in group M after the MROP, but the placenta disappeared more quickly than in group C (median; 1.0 vs. 99.0 days, *p* = 0.009). In group C, RPOC disappeared in approximately half of the cases at 100 days postpartum, and in 80% after 1 year postpartum. RPOC resolved spontaneously except in all five cases in which it was surgically removed (Figure 7).

### 3.4. Serial Changes in Serum hCG

Finally, we examined the transition in serum hCG levels in 16 cases in group C. Of the 16 cases in which serum hCG was measured, the median time from delivery to a serum hCG level below the measurable threshold was 67 (23–113) days. Next, we investigated the changes in serum hCG levels in cases in which serum hCG was examined at least twice before falling below the measurable threshold. In seven cases that delivered after 34 weeks of gestation but did not undergo UAE, the median half-life of serum hCG was 4.7 (2.3–7.5) days (Figure 8A).

Serum hCG levels decreased rapidly after UAE in three patients (Figure 8B). Among the cases who delivered before 20 weeks of gestation and did not undergo UAE, the median half-life of serum hCG was 7.4 (3.8–39.1) days (Figure 8C,D). None of the cases delivered after 34 weeks of gestation had an hCG half-life of > 10 days, whereas three cases delivered before 20 weeks of gestation had a half-life of >10 days (Figure 8D).

## 4. Discussion

If the placenta does not separate from the uterus after delivery, the NICE [5] and the WHO guidelines [6] recommend that the placenta should be removed manually within 30 min to 1 h after delivery with a set-up to deal with heavy bleeding. If bleeding does not stop, UAE or hysterectomy is required [7]. In Europe and the United States, deliveries are centralized in regional general hospitals. This allows obstetrical providers to perform MROP under anesthesia, which is performed by anesthesiologists. In Japan, on the other hand, private clinics handle half of all deliveries, and high-risk cases are transported to advanced medical facilities. As a result, many cases with RPOC have been referred to our hospital. In this study, 41 cases of RPOC delivered after 34 weeks of gestation were successfully treated without hysterectomy or death. However, fifteen of the 19 cases with MROP within 24 h of delivery experienced heavy bleeding during the procedure or on the following day. A Japanese multicenter retrospective study demonstrated that the incidence of bleeding of 1000 mL or more at delivery was significantly higher in 41.1% of the women who underwent MROP than in 4.1% of those who did not [20]. In a third stage of longer than 60 min, the incidence of postpartum hemorrhage was 70.3% with MROP and 21.2% without MROP [21]. The reason for heavy bleeding during or after MROP is considered to be that uterine contraction cannot prevent damage to large blood vessels, which occurs after MROP when decidua formation is insufficient or absent [22]. In one study, 7 of 37 MROP cases were reported to have undergone hysterectomy due to massive bleeding, and those cases were pathologically diagnosed with placenta accreta and no decidua formation [23]. Interestingly, in the current study, when MROP was performed on patients with no bleeding or other emergencies, heavy bleeding and partially retained placenta were frequently observed both intraoperatively and postoperatively (group M, Figure 2A). Regarding other complications of MROP besides heavy bleeding, the placenta was not completely removed and a retained placenta was observed in 3% of the patients [24]. Retained placenta leads to heavy bleeding and endometritis [25]. In the present study, there were five cases (26%) in which the placenta remained after MROP; the reason for this is unknown, but the frequency was much higher than previously reported. Of these five cases, two had heavy bleeding and one had infection. When obstetrical providers perform MROP, preparation for complications and detailed explanations for patients are required.

Of the 22 RPOC cases managed conservatively (group C), six cases experienced spontaneous heavy bleeding. Previous studies reported that the RPOC diameters ≥ 4–4.4 cm [9,11] or blood flow in RPOC [10,26] had been associated with heavy bleeding during conservative management of RPOC. However, these studies differ from the present study in several respects. First, these studies included cases before 20 weeks’ gestation [9,11,26]. Although the size of RPOC is smaller, trophoblast cell viability is higher in early and mid-pregnancy than in late pregnancy [27]. Furthermore, these studies included cases in which the placenta was manually removed within 24 h after delivery [9,11]. Therefore, the results of previous studies were based on a heterogenous population and may have had various biases. In the present study, there was no correlation between heavy bleeding and RPOC size or blood flow (Table 1). Further research is needed to determine the relationship between RPOC size, blood flow, and heavy bleeding.

The median time for conservatively treated RPOC to develop massive bleeding is reported to be 22–42 days, with a maximum of 38–43 days [9,11]. In this study, no events such as heavy bleeding, blood transfusion, UAE, or infection were observed after 60 days during conservative management (Figure 4). This result may help patients understand when the risk of bleeding decreases during conservative management. There were six cases in which placental removal was attempted despite the absence of heavy bleeding (Figure 2B). None of the cases resulted in the removal of the placenta; instead, the placenta spontaneously disappeared. Therefore, if conservative management does not cause bleeding or infection, placental removal should be avoided and spontaneous resolution can be expected. In this study, the rate of RPOC disappearance during conservative management was approximately 50% at 100 days and 80% at 1 year (Figure 7). In previous studies, the time to spontaneous disappearance of RPOC ranged from 48 to 84 days [9,10,28,29], which was shorter than that in the present study, likely due to the small size of RPOC in early to mid-pregnancy. This study is the first to demonstrate the time until the spontaneous resolution of RPOC in late pregnancy. More research is needed to determine whether these data can be validated under the same conditions.

The half-life of serum hCG after the intramuscular injection of hCG is 30–32 h in the absence of hCG-producing tissues in the body [30]. The longer half-life indicates that hCG-producing trophoblast cells do exist, but gradually undergo apoptosis. In RPOC cases following delivery before 22 weeks of gestation, serum hCG has been reported to be below the measurable threshold at 67 (6–183) days postpartum, and no cases experienced heavy bleeding afterwards [28]. The present study also showed that heavy bleeding did not occur when serum hCG levels fell below the measurable threshold. Therefore, it is important to measure serum hCG levels over time to assess the risk of heavy bleeding. We previously reported an average serum hCG half-life of 5.2 days in five cases of RPOC [31], similar to the average of 4.7 days in this study (Figure 8A). Interestingly, in three cases, serum hCG levels decreased rapidly after UAE (Figure 8B), suggesting that acute interception of uteroplacental blood flow causes the sudden death of trophoblast cells. There were cases of RPOC after 20 weeks’ gestation in which the half-life of hCG exceeded 10 days (Figure 8D), but there were no such cases after 34 weeks. The molecular mechanisms of apoptosis in postpartum trophoblast cells may differ between early and mid-term and late pregnancy. Methotrexate is effective in treating ectopic pregnancy and RPOC during early pregnancy [32,33]. Methotrexate has also been used for RPOC in late pregnancy [12], but its effectiveness has yet to be proven. The effect of methotrexate on RPOC in late pregnancy should be confirmed by measuring the serum hCG half-life and comparing it with the data obtained in this study.

The strengths of this retrospective study are as follows: (i) The research period was relatively short (10 years), and the participants were treated consistently. (ii) The clinical characteristics of the participants were homogeneous and were limited to cases delivered after 34 weeks of gestation. (iii) The detailed clinical course of surgical removal and spontaneous disappearance of RPOC were verified in all cases. A limitation of this study is that it was a retrospective study with a small sample size. Future randomized controlled trials are needed to evaluate whether MROP within 24 h or conservative treatment is better in RPOC without urgent symptoms.

## 5. Conclusions

The present study revealed that MROP and conservative management have their advantages and disadvantages. In MROP, the RPOC disappearance period is shorter, but hypohemoglobinemia tends to be more severe due to unexpected massive hemorrhage in an extremely short period with no time for blood transfusion. In conservative treatment, the RPOC disappearance period is longer, but hypohemoglobinemia tends to be mild due to blood transfusion preparation. Randomized controlled trials are required to determine the superiority of MROP or conservative treatment.

## Figures and Tables

**Figure 1 healthcare-11-00168-f001:**
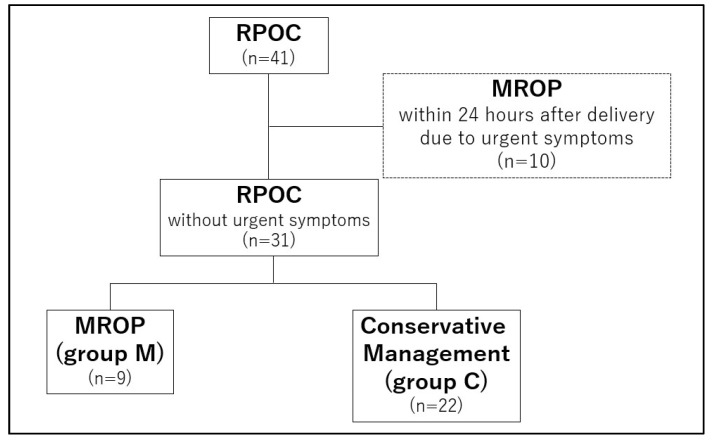
Flowchart of the present study. RPOC: retained products of conception. MROP: manual removal of placenta.

**Figure 2 healthcare-11-00168-f002:**
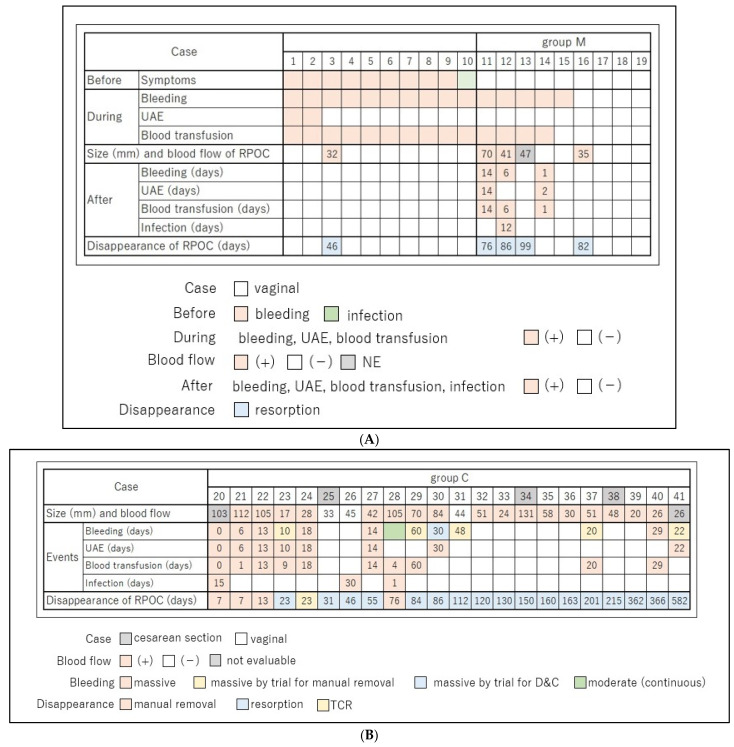
(**A**). Clinical courses of 19 cases with placental removal within 24 h after delivery. The numbers in “Size and blood flow of RPOC” indicate the size of RPOC (mm). The numbers in “After” and “Disappearance of RPOC” indicate the duration from delivery to the onset of the events (days). RPOC, retained products of conception. (**B**). Clinical courses of 22 cases with conservative management. The numbers in “Size and blood flow of RPOC” indicate the RPOC size (mm). The numbers in “Events” and “Disappearance of RPOC” indicate the duration from delivery to the onset of event (days). RPOC: retained products of conception.

**Figure 3 healthcare-11-00168-f003:**
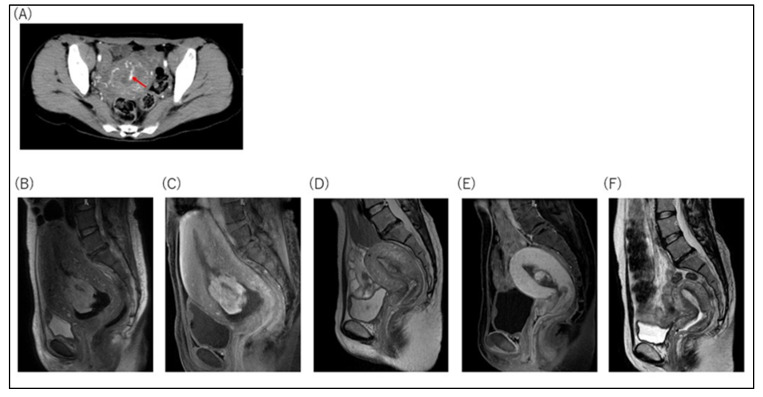
Representative images of RPOC cases with conservative management. (**A**) Case 23: Contrast-enhanced CT showed strongly contrast-enhanced RPOC in the arterial phase (postpartum day 10). Arrow shows RPOC. (**B**–**F**) Case 37: Serial changes in contrast-enhanced MRI during conservative management. (**B**) Sagittal T2WI showed an RPOC in the uterus. The RPOC size was 72 × 42 × 49 mm (postpartum day 3). (**C**) Sagittal contrast T1WI demonstrated strongly contrasted RPOC (postpartum day 3). (**D**) Sagittal T2WI showed RPOC in the uterus, whose size became smaller. The RPOC size was 22 × 17 × 15 mm (postpartum day 22). (**E**) Sagittal contrast T1WI still demonstrated strongly contrasted RPOC (postpartum day 22). (**F**) MRI did not detect RPOC (postpartum day 64). Ultrasound showed RPOC of 8 mm on postpartum day 87. On postpartum day 102, hysteroscopy showed that the RPOC was 10 mm, and on postpartum day 201, it was finally confirmed that the RPOC had disappeared. CT: computed tomography; RPOC: retained products of conception; MRI: magnetic resonance imaging; T2WI: T2-weighted imaging; T1WI: T1-weighted imaging.

**Figure 4 healthcare-11-00168-f004:**
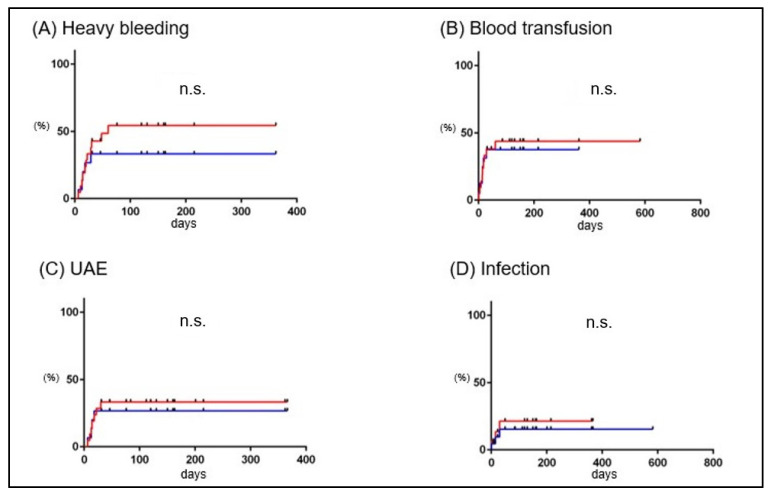
The cumulative incidence rate of events in group C (RPOC cases with conservative management). The horizontal and vertical axes represent the duration from delivery to the onset of the events (days) and the incidence rate (%), respectively. Red lines and blue lines show cumulative incidence rates of heavy bleeding (**A**), blood transfusion (**B**), UAE (**C**), and infection (**D**) in 22 cases with conservative management of RPOC and in 16 cases without placental extraction trial during conservative management. RPOC: retained products of conception; UAE: uterine artery embolization; D&C: dilatation and curettage.

**Figure 5 healthcare-11-00168-f005:**
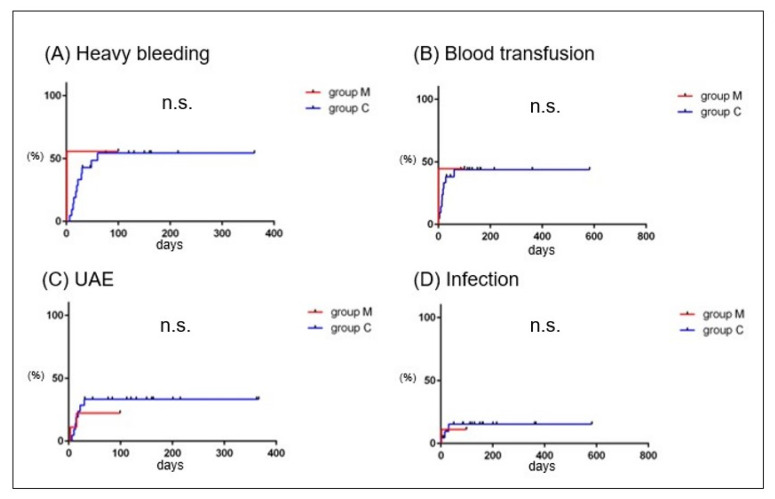
The cumulative incidence rate of events in Group M and Group C. The horizontal and vertical axes represent the duration from delivery to the onset of the events (days) and the incidence rate (%), respectively. Red lines and blue lines show cumulative incidence rates of heavy bleeding (**A**), blood transfusion (**B**), UAE (**C**), and infection (**D**) in Group M (*n* = 9) and Group C (*n* = 22). RPOC: retained products of conception; UAE: uterine artery embolization.

**Figure 6 healthcare-11-00168-f006:**
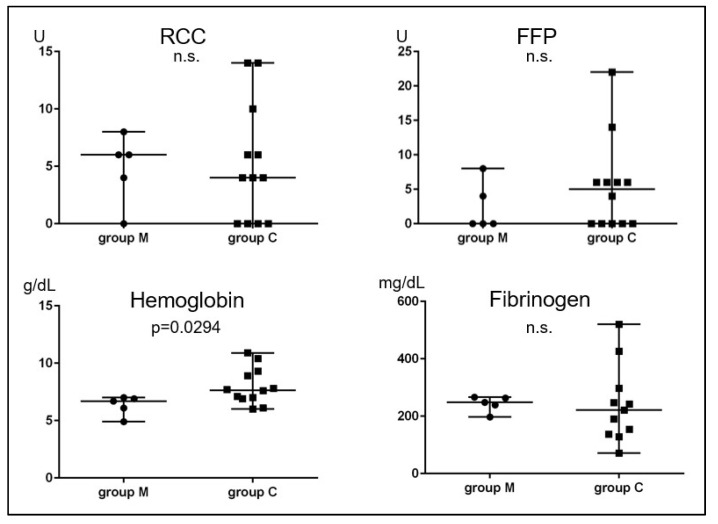
Univariate analysis between group M and group C at heavy bleeding. RCC: red cell concentrate; FFP: fresh frozen plasma.

**Figure 7 healthcare-11-00168-f007:**
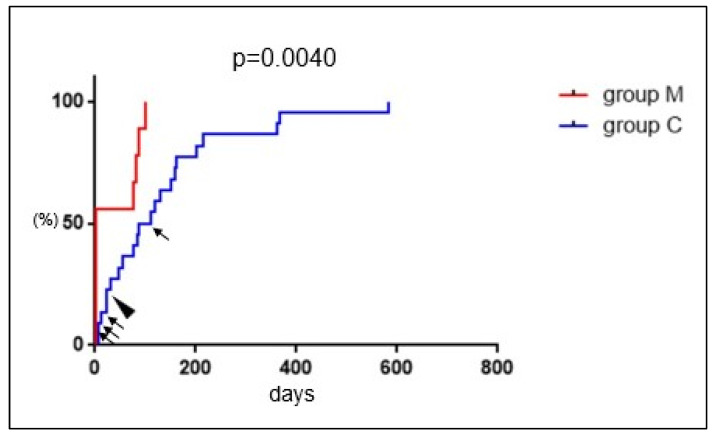
The cumulative rate of RPOC resolution. The horizontal and vertical axes indicate the duration from delivery to resolution (days) and the cumulative incidence rate (%), respectively. Arrows and arrow head indicate the timing of MROP and TCR, respectively. RPOC: retained products of conception; MROP: manual removal of the placenta; TCR: transcervical resection.

**Figure 8 healthcare-11-00168-f008:**
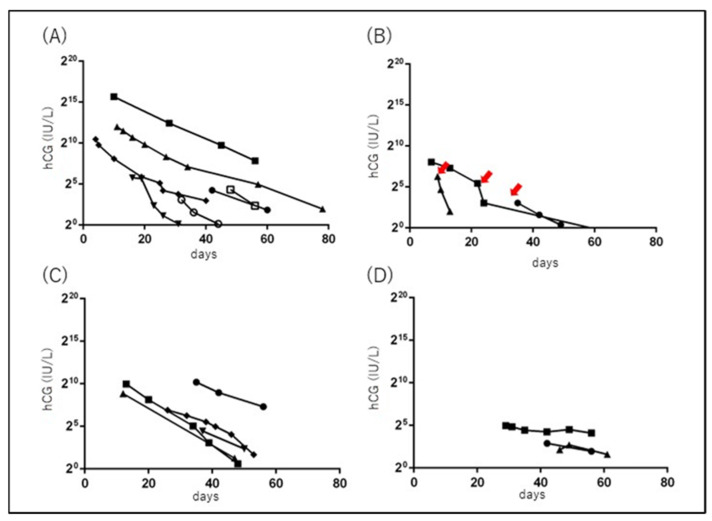
Serial changes in serum hCG. The horizontal and vertical axes indicate the duration from delivery to the measurement date (days) and the serum hCG level (IU/L), respectively. (**A**,**B**) RPOC cases delivered after 34 weeks of gestation. (**A**) Cases without UAE. (**B**) Cases with UAE. Arrows show the date on which UAE was performed. (**C**,**D**) RPOC cases delivered at less than 20 weeks of gestation and did not undergo UAE. (**C**) Cases with serum hCG half-life of less than 10 days. (**D**) Cases with serum hCG half-life of more than 10 days. RPOC: retained products of conception; UAE: uterine artery embolization; hCG: human chorionic gonadotropin.

**Table 1 healthcare-11-00168-t001:** Univariate analysis between RPOC with heavy bleeding and light bleeding during conservative management. RPOC: retained products of conception.

	Heavy Bleeding (n = 12)	Light Bleeding (n = 9)	*p* Value
Gestational age at delivery (weeks), median (range)	39.9 (35.1–41.4)	38.1 (36.0–41.1)	0.2881
Mode of delivery, n (%)			
vaginal delivery	12 (100)	6 (66.7)	0.0632
cesarean section	0 (0)	3 (33.3)	
RPOC length (mm) at diagnosis, median (range)	47.5 (17.0–112.0)	45.0 (20.0–131)	0.6511
RPOC with flow on MRI or CT, n (%)	9/12 (75.0)	8/9 (88.9)	1

## Data Availability

The data that support the findings of this study are available from the corresponding author.

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
