# Peer review of "Conservative Management for Retained Products of Conception in Late Pregnancy"

_healthcare, 2023, doi:10.3390/healthcare11020168_

Round 1

Reviewer 1 Report

This is an interesting paper and a valuable topic to be published. Reads like paper was written by different people doing different section. Strongly suggest that one person undertakes thorough edit to make this a more consistent read.

Title does not reflect content so should be clearer

Abstract would benefit from being presented in research format - aim, background, results, discussion, conclusion.

Too much on the results in here that are not clearly presented at all. Should include conclusion and recommendations.

line 10-12 - saying the same twice with slight variation. Make the second sentence - this was achieved through retrospective analyis - or something like that.

Line 13 - start by saying - of the 41 cases ... as this is not clear at all. Use numbers not writing out as this makes it harder to follow and most of the paper you have used numbers - be consistent.

Define terms - such as retained products

Line 15 - not clear

Results in abstract need to be more general and not so specific.

Line 21 - suddenly discuss UAE with no explanation or context

Line 28 need reference

Line 37 define conservative management

Line 39 sentence not clear

Line 53 sentence should go in discussion/conclusion

Line 61 should be mentioned earlier if included. Explain UAE and why used. Do not assume reader understands what you are writing about. Explain points fully

Line 68 blood flow in RPOC ? means there was doppler studies undertaken - needs explaining/defining

Line 81 should have mentioned conservative management earlier and explain what this is - provides a context to the study

Line 83/5 should be earlier and in abstract as it clearly explains this

Line 96 not clear which group the 2 cases belong to

Line 98 14 cases of what, same with 13 cases. This is all very hard to follow as it is clearly presented or explained. Do not assume the reader understands. Also need to refer to figure and describe what is in the figure.

Line 107 3 cases of what, section is not clear. Explain use of UAE and MRI - was this done with all cases or who

Line 130 should be twelve not 12 for start of sentence

Explain more as not clear

Line 148 in which cases does this refer to

Line 172 how many cases

Line 201 discuss results more before putting in the strengths whihc usually go at the end of study. 

Were there any limitations

What were the recommendations from this study.

Author Response

Thank you very much for reviewing our manuscript and offering valuable advice. We have addressed your comments with point-by-point responses, and revised the manuscript accordingly. Please see the attachment.

Reviewer 2 Report

The clinical problem is really important. The question often is "to do or to wait"? This means that such studies are highly needed to improve our treatment abilities. In revised paper the Authors presented group of 41 patients, collected data and compared some results, but I have serious problems with analyzing/reading this paper. 

1. I would expect reformulation of submitted paper. It should be reorganized. The aim is visible -> comparison, but it wasn't clearly presented in which aspects. And what is more important there are no clear conclusions on the basis of particular results (pointed out). In the conclusions part the Authors speculate and additionally present study limitations... 

2. In the methodology part I feel that comparison of >34 with >37 and 40/41 hbd together is not very good idea. For sure it is more important, and cannot be proposed in the neonatological perspective. But we also know that uterus looks and reacts completly different in this terms/groups... If you don't have enough cases it has to be also highlighted in the limitations. 

3. I cannot agree to quantification of blood loss defined as presented... Even if it was somehow (semi-quantitavely counted) frames sould be given...

4. Figures are totally non-informative. Figure 1A and 1B - descriptions are difficult to read, first 4 rows are useless if all are empty... Table 1, why only heavy and light were compared? Figure 3 why A and E, B and F, etc. were not shown together with different colours to provide easier comparison? 

5. Were there any attempts of histeroscopic evacuation or coagulation of blood vessels in perspective of mentioned D&C?  

5. References should be refreshed. There are some really old papers, it would improve value of the manuscript. 

A manuscript worth publishing but after thorough corrections, mainly technical ones.

Author Response

We wish to express our appreciation to your insightful comments on our paper. The comments have helped us improve the paper. We have addressed your comments with point-by-point responses, and revised the manuscript accordingly. Please see the attachment. 

Reviewer 3 Report

This is a retrospective study enrolled 41 cases with retained products of conception (RPOC) delivered after 34 weeks of gestation to analyze the clinical course of RPOC and elucidate its natural history. The results of this study indicate that conservative management can be considered as a standard treatment for RPOC. However, there are still some limitations in this study:

1. The sample size of this study was too small to draw the conclusions;

2. There are some mistakes in the format of subtitles in the line 86 and 106. The first number in the title should be written in English. The same error was also found in line 130;

3. A flowchart of this study is recommended to add in the current study for readers to understand easily;

4. There is an incorrect place in line 92. It should be “six of 15 cases who underwent MROP without bleeding experienced heavy bleeding during the procedure (cases 10-15)”;

5. In figure 3, RPOC cases with manual removal of the placenta (MROP) group was recommended to add in the analyses of the cumulative incidence rate. The log-rank test can be applied to illustrate the difference between two groups;

6. This study only present the situations of 41 cases with RPOC. More statistical methods should be applied, such as logistic regression or constructing AI models, to help clinicians making decisions on surgical treatment or conservative management;

7. The part of Discussion was too simple for the current study. The previous studies about the influence factors in MROP and the significance for clinicians of this study should be added into Discussion.

Author Response

We really appreciate your efforts in reviewing our manuscript. We feel your comments have helped us improve the paper. We have revised the manuscript accordingly. Please see the attachment.

Round 2

Reviewer 2 Report

Thank you for all explanation and amendments introduced into your manuscript, it clarified me a lot and the paper is much more friendly for the readers. I don't have any further remarks. I would recommend collection of the data for prospective analysis, also taking into account elements which were missed in revised paper due to retrospective character. 

I recommend for publication. 

Author Response

We appreciate all of your insightful comments. We plan to collect data for prospective analysis as you recomended. Thank you for taking the time and energy to help us improve the paper. 

Reviewer 3 Report

The current study was revised point by point, and can be accepted for publication after correction to minor spelling and grammar mistakes.

Author Response

Thank you for giving us the opportunity to strengthen our manuscript with your valuable comments and queries. We are grateful for the time and energy you expended on our behalf. We have corrected spelling and grammar mistakes.